# The Best Peptidomimetic Strategies to Undercover Antibacterial Peptides

**DOI:** 10.3390/ijms21197349

**Published:** 2020-10-05

**Authors:** Joanna Izabela Lachowicz, Kacper Szczepski, Alessandra Scano, Cinzia Casu, Sara Fais, Germano Orrù, Barbara Pisano, Monica Piras, Mariusz Jaremko

**Affiliations:** 1Department of Medical Sciences and Public Health, University of Cagliari, Cittadella Universitaria, 09042 Monserrato, Italy; barbara.pisano@unica.it (B.P.); monipiras@hotmail.com (M.P.); 2Division of Biological and Environmental Sciences and Engineering (BESE), King Abdullah University of Science and Technology (KAUST), Thuwal 23955-6900, Saudi Arabia; kacper.szczepski@kaust.edu.sa; 3Department of Surgical Science, OBL Oral Biotechnology Laboratory, University of Cagliari, 09124 Cagliari, Italy; alessandrascano@libero.it (A.S.); ginzia.85@hotmail.it (C.C.); sarafais79@gmail.com (S.F.); orru@unica.it (G.O.)

**Keywords:** antimicrobial peptides, antibiotics, peptidomimetics, polymers, metal coplexes

## Abstract

Health-care systems that develop rapidly and efficiently may increase the lifespan of humans. Nevertheless, the older population is more fragile, and is at an increased risk of disease development. A concurrently growing number of surgeries and transplantations have caused antibiotics to be used much more frequently, and for much longer periods of time, which in turn increases microbial resistance. In 1945, Fleming warned against the abuse of antibiotics in his Nobel lecture: “The time may come when penicillin can be bought by anyone in the shops. Then there is the danger that the ignorant man may easily underdose himself and by exposing his microbes to non-lethal quantities of the drug make them resistant”. After 70 years, we are witnessing the fulfilment of Fleming’s prophecy, as more than 700,000 people die each year due to drug-resistant diseases. Naturally occurring antimicrobial peptides protect all living matter against bacteria, and now different peptidomimetic strategies to engineer innovative antibiotics are being developed to defend humans against bacterial infections.

## 1. Introduction

Antimicrobial peptides (AMPs), also known as host defense peptides (HDPs), are produced by all living matter as a critical part of the innate immune system [1,2,3]. Their existence was discovered in 1939, the year gramicidin was isolated from the bacteria, *Bacillus brevis*; some resources, however, claim that the discovery of lysozyme in the 1920s should be treated as the first AMP instance, due to lysozyme’s non enzymatic, bactericidal second mode of action [2]. As of 2019, over 3000 AMPs have been isolated across all kingdoms [4]. Most of them have variable sequence length—from 10 up to 60 amino acid residues, mainly in L-configuration [4]. AMPs are expressed in many cell types in response to the activation of the Toll-like receptor (TLR) signaling pathway [1]. They are predominantly comprised of two types of amino acids residues—cationic residues such as Arg, Lys, and His, and hydrophobic residues (mainly aliphatic and aromatic) [1]. Both cationic and hydrophobic residues engage in the antimicrobial mechanism of action. Furthermore, AMPs can be classified into three groups, based on their structures (Scheme 1): α-helical, β-sheets, and extended peptides. α-helical peptides are the largest group of AMPs with characteristic qualities such as amphipathic properties, and the ability to possess a tertiary structure with a hinge in the middle of a chain [1,4]. β-sheets peptides feature one to five disulfide bridges that help to stabilize their bioactive conformation. Peptides rich in amino acids such as proline, arginine, tryptophan or glycine usually have a linear structure [1,4].

HDPs have a wide range of antibacterial activities, and low susceptibility to antimicrobial resistance [5,6,7,8,9,10,11,12]. Nevertheless, the use of HDPs is limited due to their low resistance to proteases, and their increased cost of preparation [8,13,14]. Different peptides [15,16,17,18], peptoids [19,20,21,22,23], polymethacrylate derivatives [24], polynorbornene derivatives [25], polycarbonate derivatives [26], peptide polymers [27,28,29,30,31,32], and polymers made by controlled living radical polymerization (CLRP) [33] were developed to mimic the antibacterial properties of HDPs, and to ameliorate their shortcomings [34].

The widely developed peptide-drug research has led to the introduction of around 60 FDA approved peptide therapeutics in the market [35]. Over 140 are in clinical trials, and over 500 are in the preclinical stages. Although numerous biomimetic and de novo designed AMPs are at different stages of clinical trials [36,37], only eight AMPs received FDA approval, i.e., daptomycin (approved in 2003) and oritavancin (approved in 2014) [37].

In this review paper, we describe innovative peptidomimetics strategies and we present successful examples of new antimicrobial peptide-mimicking antibiotics. The aim of this review is to gather the latest advances in the research of new antibiotics for those who are engineering future antibacterial strategies.

## 2. Samsons Hair of AMPs

There are three main factors—charge, hydrophobicity and amphipathicity, which determine the activity of AMPs, and they are all related to their intrinsic properties.

The positive charge of AMPs (between 2+ and 9+) is one of the most essential factors responsible for their antibacterial activity [38]. Positively charged AMPs outcompete the binding of native Mg(II) and Ca(II) ions to lipopolysaccharides (LPS), and destabilize the outer membrane of Gram-negative bacteria [39]. Consequently, the destabilized regions enable the peptide to pass into the cell. Once through, AMPs bind to the cytoplasmic membrane and cause depolarization and pore formation, resulting in cell death [40]. In the case of Gram-positive bacteria, AMPs must overcome the outer wall with two main obstacles, peptidoglycan and teichoic acids, before an AMP interacts with the cytoplasmic membrane. AMPs overcome peptidoglycan because it is able to penetrate through the relatively porous peptidoglycan. The porous nature of peptidoglycan makes it easy for small molecules (< 50 kDa) to penetrate through it, and furthermore, it does not possess a negative charge, which could prevent the penetration of positively charged AMPs through it [40,41,42]. To overcome the anionic teichoic acids, they can either act as a ladder that will help positively charged AMPs to travel to the cytoplasmic membrane, or the can act as “cages” that entrap AMPs inside the bacteria, thus reducing their local concentration on the membrane The behaviors of teichoic acids vary based on the type of AMP and the type of bacteria [42]. After crossing the outer wall, AMPs can disrupt the integrity of the cytoplasmic membrane, leading to membrane disruption and dislocation of peripheral membrane proteins [42].

The hydrophobicity of AMPs is another important factor. It is assumed that hydrophobic residues help to insert peptides into the bacterial membrane and further impair the membrane [4,42]. It was also revealed that hydrophobic residues account for 40–60% of all amino acids in AMPs [38]. Moreover, the hydrophobicity is strongly correlated with antimicrobial activity, where increasing the hydrophobicity to a certain level improves antimicrobial activity [1,38]. The relationship between hydrophobicity and antimicrobial activity is also proven to be the case for peptidomimetics such as peptoids, where amino acid side chains are linked to the peptide backbone through the amide nitrogen rather than to the α-carbons [43]. However, a higher hydrophobicity also increases hemolytic activities, resulting in unwanted toxicity to eukaryotic cells [1,38]. Studies with model peptides have shown that a continuous region of 4-6 hydrophobic residues is sufficient to sustain the antimicrobial activity of peptides, while simultaneously reducing the hemolytic activity [44].

The amphipathicity of AMPs, created by the segregation of hydrophobic and polar residues on the opposite sites of the backbone, is another factor related to their activity [1]. Amphipathicity is considered the strongest indicator of AMP activity, and is described by the hydrophobic moment, defined as the vector sum of the hydrophobicity of each amino acid located in the helix [1,45]. The mechanism by, which amphipathicity works is the result of previous factors. The positively charged regions of AMPs enable binding to the anionic phospholipid head groups of the bacterial membrane. Meanwhile, the hydrophobic regions of peptides invade the lipid bilayer and interact with the hydrophobic acyl chains of phospholipids, which results in membrane penetration [38]. Several studies have shown the importance of amphipathicity in the process of AMPs binding to membranes, and the negative impact on activity against Gram-positive bacteria and fungi, when amphipathic character of the peptide was eliminated [13,46,47,48].

## 3. AMPs: Achilles’ Heel

Despite the recent advancements in AMP research, there are still major challenges to overcome. One of the main obstacles in AMP applications is the proteolytic instability of peptide drugs [49,50]. Peptides are targeted by numerous proteolytic enzymes located in bodily fluids and tissues of the host. Moreover, AMPs can be recognized as an antigen and targeted by host immune system [51,52]. One of the countermeasures for their potential immunogenicity could be glycosylating AMPs with a glycosyl profile similar to that of the host. For instance, covering AMPs with polysialic acid (derived from *Neisseria meningitidis* or *E. coli* that shares structural mimicry with host cell lectins), or its analogues, can lessen the immunological response [52]. Nevertheless, some bacteria have already developed resistance to AMPs. For example, LL-37, a human antimicrobial peptide, can be digested by proteases synthesized by *P. aeruginosa*, *Enterococcus faecalis*, *Proteus mirabilis*, *S. aureus*, and *S. Pyogenes* [50,53]. Synthesizing proteases to target specific antimicrobial peptides is one of the many ways bacteria participate in this evolutionary “arms race”. Other methods used by bacteria to cope with AMPs, including the use of efflux pumps, lowering the binding affinity, or reducing the anionic charge of lipopolysaccharides on their surface, are exhaustively discussed in [50,54].

The high toxicity of AMPs in eukaryotic cells is another one of their shortcomings. This high toxicity can lead to hemolysis, nephrotoxicity, and neurotoxicity [1,55,56,57]. More studies need to focus on the pharmacokinetics and pharmacodynamics of AMPs in order to determine a proper drug dosage that will maintain a balance between positive and negative outcomes [2].

The bioavailability of AMPs is rather low. Peptides are hardly absorbed by the intestinal mucosa, and their pharmacological distribution requires additional financial input. In recent years, much progress has been made in this field—e.g., nano-carriers not only increase the bioavailability, but also lower cytotoxicity and reduce degradation of a compound, leading to increased efficiency [37]. Despite different efforts, the entire process of AMPs design and discovery (costs of synthesis and screening) is relatively high and does not guarantee the expected outcome [58]. The high production cost of AMPs (as for 2006, producing 1 g of peptide using solid-phase peptide synthesis (SPPS) costs USD 100–600) limits the development and testing of new AMPs, even if recent improvements in SPPS lowered the costs of synthesis [8].

AMPs, as other peptide drugs, can induce immune responses and cause allergies [59]. Immunogenicity depends on the host immune system, but is also correlated with the peptide’s dose, duration and frequency of treatment, route of administration, and the patient’s pathologies [60,61,62,63]. Immunogenic response can affect the peptide-drug efficiency and even lead to allergy and/or hypersensitivity [63]. Bering that in mind, it is important to study the immunogenic and allergic activity of AMPs. In vitro tests together with mathematical calculations of allergens similarities, with the use of specific databases of allergenic compounds [64], i.e., http://www.allergome.org, can give nowadays reliable results [65].

## 4. Bacterial Membranes vs. Human Cell Membranes

AMPs’ selectivity depends on the intrinsic differences in the cell wall of prokaryotes and eukaryotes [66]. Since most bacterial plasma membranes are rich in anionic phospholipids (such as phosphatidyl-glycerol (PG), cardiolipin (CL), and phosphatidylserine (PS)) and since the outer monolayers of eukaryotic membranes are composed of zwitterionic (overall neutral) lipids, the prokaryotic membranes are more electronegative than eukaryotic membranes [66]. AMPs’ selectivity towards either zwitterionic or negatively charged lipids is crucial in determining their bioactivity, toxicity, and potential use as drugs [67]. Furthermore, in some cases, the high electronegative nature of a membrane helps AMPs to adopt active conformations and interact with the membrane.

The cell-walls of Gram-positive and Gram-negative bacteria vary significantly. In Gram-positive bacteria, the cell wall is comprised of multiple layers of peptidoglycan composed of a poly-[*N*-acetylglucosamine-*N*-acetylmuramic acid] backbone, with a thickness of 30–100 nm [68]. The peptidoglycan wall determines the shape of bacteria, prevents cell lysis (due to high internal osmotic pressure), and defends the cell from environmental hazards such as antibiotics. The cell wall is interspersed by two types of anionic polymers—teichoic acids, which are linked to peptidoglycan, and lipoteichoic acids that are anchored to the cell membrane. Anionic polymers play an important role in ion homeostasis, regulate cell morphology, division and autolytic activity, and protect bacteria against the host’s defense mechanisms and antibiotics [69]. When compared to Gram-negative bacteria, Gram-positive bacteria have a much higher content of negatively charged lipids, predominantly phosphatidylglycerol (PG) and cardiolipin (CL) [66].

The envelope of Gram-negative bacteria can be divided into three layers: the outer membrane (OM), the peptidoglycan cell-wall, and the cytoplasmic or inner membrane (IM). The outer membrane is composed mainly of glycolipids, and principally, of lipopolysaccharides (LPS). LPS are composed of three domains: lipid A, the core oligosaccharide, and the O-antigen [70]. LPS create a permeability barrier and protect the bacterial cell against various environmental factors such as antibiotics and bile salts. However, the LPS are primary bacterial components encountered in the host immune system and play an important role in bacterial pathogenicity (the host organism can treat them as a pathogen-associated molecular pattern; PAMP). The inhibition of LPS biosynthesis could also trigger stress response pathways and affect the assembly and function of membrane proteins [71]. In addition to LPS, the outer membrane contains lipoproteins and β-barrel proteins that can act as gated channels for vitamin transport. In Gram-negative bacteria, the peptidoglycan cell-wall plays the same role as in Gram-positive bacteria, but it is thinner—ranging from 2.5 to 7 nm (1–3 layers of peptidoglycan) [72]. The inner membrane of bacteria serves as a platform for all membrane-associated functions of eukaryotic organelles, which include energy production, lipid biosynthesis, protein secretion, and the transport of molecules [73].

Despite all the differences in the cell wall structure, there are three models, which describe how AMPs destabilize the bacterial membrane (Table 1). In the barrel-stave model, a variable number of peptides are inserted perpendicularly into the bilayer, forming a barrel-like ring with a central lumen acting as a pore [74]. In the carpet model, peptides are positioned parallel to the bilayer, and form carpet-like structure on the membrane surface. After reaching a concentration threshold of adsorbed AMP, accumulated AMPs change membrane fluidity and reduce the barrier properties of the membrane, producing a “detergent-like” effect (formation of micelles), and resulting in membrane disruption [74]. In the last model, called toroidal-pore model, the peptides are inserted perpendicularly into the bilayer and induce a bend in the membrane. This results in a formation of a pore that constitutes partially peptides and partially phospholipid head groups. The difference between toroidal pores and barrel-stave pores is that peptides are always intercalated with the lipids head groups when forming a pore.

Although, AMPs’ main antimicrobial activity is membrane disruption, several studies have shown that antimicrobial peptides exhibit other mechanisms of action. AMPs can inhibit the synthesis of cell-walls, nucleic acids and proteins, and they can inhibit cell division. Moreover, AMPs are able to target organelles of eukaryotic pathogens [81]. Many of these aspects are broadly described in the literature [82]. For instance, recent results showed that IARR-Anal10, the analog of mBjAMP1, exhibits no effect on the permeability or integrity of the bacterial outer membrane, and acts intracellularly rather than at the cell membrane. Moreover, IARR-Anal10 can bind bacterial DNA [82].

## 5. Antibacterial Peptides in the Management of Oral Infections with Biofilm Production

The oral cavity is unique in the level of microbial diversity it holds, and it harbors about 1000 microbial species. Most of them have physiological functions in oral tissue, and the majority of oral microbes are considered commensals [83]. Others are associated with oral diseases and can be divided into two, ideal groups: “always” pathogenic and “potentially” pathogenic.

A pathogen is a microorganism capable of causing disease in healthy individuals. It can enter the host organism, establish a niche within which to replicate, and disseminate to reach a new host. It is now possible to identify different genes associated with pathogenicity factors inside the pathogen genome. Compounds that harm the host are naturally expressed, and they are not associated with stress conditions in the oral cavity.

An opportunistic pathogen does not cause disease in healthy individuals, but it can cause pathological states in “at-risk” patients, for instance, during a dysbiosis status in the oral tissues. In this context, an opportunistic bacterium could be useful or even essential during human life, but it can also become a pathogen under different chemical and biological stimuli released by the host’s tissues [84,85,86].

Periodontitis and periimplantitis are two of the most pervasive pathologies in dental medicine. According to recent publications and WHO reports, millions of peoples in the industrialized countries are toothless, and over hundred-and-twenty million are missing at least one tooth. Moreover, from 70 to 90% of people over 60 years old have dentures. Periodontitis is an infectious disease caused by microbes, the majority of which is represented by Gram-negative anaerobic rods [8], capable of harming periodontal tissues by direct products (e.g., proteases), and mark an unusual inflammatory response, which leads to the destruction of the underlying tissue and tooth loss in adults [9].

In the same way, peri-implantitis (PI) is a specific infective-inflammatory site-specific disease that occurs in dental osseointegrated implants. The onset of this infection is represented by the adherence and proliferation of the periodontal anaerobic bacteria on implant surfaces and it is strongly linked to an inflammatory process that occurs and may be limited to the soft-tissues, or is associated with the progressive loss of osseointegration, mucositis, and peri-implantitis, respectively. Peri-implantitis shows a remarkable impact on health status and related costs in industrialized countries. Recent reports indicate that the percentages of clinically evident PI range from 30 to 60%, and different efforts are undertaken to treat this infectious disease [10].

Nowadays, periodontal therapy follows two main procedures—the with or without surgery approach. The first one is used to improve access to the root surface; nevertheless, is not effective if bacteria invade periodontal tissues [11,12]. In other cases, adjunctive systemic antimicrobial therapy remains the treatment of choice [13]. However, factors such as the mode of antimicrobial action, the susceptibility of periodontal pathogens, the dosage, correct management and use in the treatment of periodontal disease, as well as the mechanism of bacterial resistance to each antimicrobial, are still discussed among dentists and microbiologists [14].

Antimicrobial resistance is a natural phenomenon of bacterial survival, but at the same time, it represents a worldwide public health risk, particularly in patients who have chronic periodontitis or periimplantitis, and who are frequently infected with multidrug-resistant strains. For instance, *Prevotella intermedia* isolated from oral cavity has shown resistance to clindamycin, amoxicillin, doxycycline, or metronidazole [15]. In a clinical investigation performed by Ram et al., about 70% of peri-implantitis patients exhibited strains that are drug-resistant in vitro to one or more of the tested antibiotics [16]. In this context, new approaches are required to combat the spread of drug-resistant bacteria in the oral cavity, especially given that oral microbes are also associated to degenerative systemic diseases in other tissues [17].

Oral cavity microorganisms are present either as planktonic cells in saliva, or as sessile and incorporated into a complex biological matrix called a biofilm. In the mouth, the biofilms represent the conditions of severe oral pathologies such as dental caries, periodontal diseases, and periimplantitis.

The resident bacteria cells in the biofilm are encased in an exopolymeric complex matrix with amyloid-like properties. Biofilm is comprised of exopolysaccharides, lipids, proteins, and cDNA. This extracellular matrix provides mechanical rigidity and protects bacteria from the external environment, often acting as waterproof barrier (Scheme 2). The resident bacteria strictly regulate the biofilm functions via a coordinated gene expression network, including growth direction, metabolic pathways, and pathogenic compounds for the host. In this way, biofilm represent an exceptional survival strategy for pathogens, and together with corresponding changes in gene expression, can protect the bacteria from disinfectant agents or antibiotics, and from the immunity system. On the other hand, biofilm sessile conditions allow bacteria to adhere to the oral tissues e.g., teeth surface, in opposing salivary flux.

In the oral cavity, we can observe diverse biofilms located in different tissues/organs: the tongue, a dental apparatus, or in prostheses i.e., implants and orthodontic appliances. These bacterial complexes possess great phenotypic adaptability, genetic resistance, and as already discussed, are resistant to antimicrobial treatment. It is often the extracellular biofilm matrix that physically restricts the diffusion of antimicrobial agents, even if does not seem to be a predominant mechanism of antimicrobial resistance.

The oral infective diseases are host–pathogen interactions with complicated biochemical networks, and in the last decade they have been uncovered with metagenomic and proteomic analyses. Surprisingly, the resident oral bacteria can assist the pathogens, or they interact by turning opportunistic bacterium into the pathogen phenotype by a wide variety of mechanisms, such as evasion of the immune response, cell–cell signaling, metabolic interactions, etc. [87].

Inside biofilms, bacterial cells use a communication pathway called quorum sensing (QS) to coordinate the population density and growth direction. In oral bacteria, QS also modulates virulence and pathogenic functions [88], and studies on QS signaling molecules to produce a “quorum-quenching” effect that could lead to bacterial communication interference [89], and new modes of antibacterial action.

Recent studies have better defined the functions of antibiofilm peptides [26,27] (Scheme 3). Clinical studies examined the expression of AMPs in oral tissues and in crevicular fluid or saliva and correlated them with the clinical index of periodontal diseases and related microbiological analysis pathogens. Among different AMPs, Cathelicidin, α- and β-defensins 1–3 were strongly linked to periodontal status [90,91].

In the oral cavity, human AMPs are small cationic peptides, synthesized in the oral epithelium and salivary glands, and serve as defensive tools. After a look through the antimicrobial peptides databases (APD) [92], over 40 different salivary AMPs were found. These peptides are, on average, 43 amino acids long, with a medium net charge of 4.83. Oral AMPs include peptides of different biological roles: Defensins [93], Histatins, Cathelicidins, Adrenomedullin, Statherin, C-C Chemokine, Azurocidin, and Neuropeptides [94,95].

Oral peptides have evolved together with the oral microbiota and have acquired particular properties. Some AMPs families can both kill periodontal bacteria and neutralize the lipopolysaccharide (LPS) of Gram-negative bacteria. Cationic AMPs, as adrenomedullin, α-defensins (HNP), β-defensins, cathelicidin, ll-37, histatins 1 and 3, statherin, C–C motif chemokine 28 (CCL28), Azurocidin, are protected against bacterial proteases (i.e., *P. gingivalis* gingipains) by salivary proteins [79]. Other AMPs are involved in the bacterial agglutination, small salivary mucin-7 (MUC7) or can inhibit bacterial growth by acting as divalent cation scavengers (ion chelating agents) [96]. In the periodontal aetiologic process, the AMPs mechanism of inhibition of bacterial proteases, MPs hBD-2 and CCL20, or the peroxidase activity is particularly interesting. In this process, different active compounds against *A. actinomycetemcomitans*, *P. gingivalis* and oral *streptococci* [97,98] are produced.

Dysregulation of AMPs in the oral tissues is related to the pathogenesis of periodontal diseases. This aspect is observed in the patients with systemic diseases (rheumatoid arthritis, diabetes mellitus), obesity, and tobacco smokers, all of which are at high risk of periodontal pathologies. It is suggested that AMPs may act as a putative factor in these mentioned conditions and periodontal diseases.

Even if numerous in vitro experiments have shown AMPs activities versus oral pathogens bacteria, it is not clear if the AMPs exert direct antibacterial activity The concentration of AMPs in the crevicular fluid is lower than the MIC value, and some authors suggest that the primary role of oral AMPs is not linked to antibacterial activity, but inversely may serve to maintain oral microbiota homeostasis. Future solutions could be antimicrobial peptides cocktails that combat a subset of oral pathogens by modulating commensals/pathogens and by avoiding oral dysbiosis [99,100,101,102,103,104,105,106].

## 6. Mimicking Amino Acid Sequence

The inspiration for the AMP-mimicking peptides comes mostly from nature. Temporin-SHf (FFFLSRIF; Figure 1) is a short (8 amino acids long) unstructured [107] and highly hydrophobic peptide found in the frog’s skin, with moderate antimicrobial activity (MIC*^E.coli^*(μM) = 25–30; MIC*^S.aureus^*(μM) = 12.5) [107]. The tuning of its antimicrobial properties has led to the identification of an analogue peptide, TetraF2W-RR (WWWLRRIW; Figure 1), with enhanced antimicrobial (MIC*^E.coli^*(μM) = 25; MIC*^S.aureus^*(μM) = 1.6–3.1) and antibiofilm activity against methicillin-resistant *Staphylococcus aureus* (MRSA) in free and immobilized forms. Substituting phenylalanine residues with tryptophan, enhanced the antimicrobial activity, particularly against MRSA (MIC(μM) = 1.6–3.1) [108]. Horine (WWWLRRRW) and Verine-L (RRRWWWWL) are the ultimate uprade of Temporine (Figure 1), and have also enhanced the antimicrobial activity against *S. aureus* (MIC^Horine^ = 4 μM, MIC^Verine-L^ = 4 μM) and *E. coli* (MIC^Horine^ = 32 μM, MIC^Verine-L^ = 4 μM) [109].

Frogs, such as *Rana temporaria* and *Litoria aurea*, secrete numerous closely related antimicrobial peptides as an effective chemical dermal defense [110]. Short peptides, isolated from frogs of the *Rana genus (Anura)*, have intriguing antimicrobial activity against pathogenic microorganisms, and even drug-resistant strains. Each peptide has a heptapeptide loop, and reduction of the C-terminal disulfide bridge maintains the loop-like conformation and has no significant effect on hemolytic and antimicrobial activity [111]. In contrast, the substitution of the C-terminal cysteines and disulfide bridge by metathesized vinylic allylglycine residues resulted in increased antimicrobial potency [112]. Brevinin-1BYa (FLPILASLAAKFGPKLFCLVTKKC) was first isolated from skin secretions of the foothill yellow-legged frog *Rana boylii*, and showed broad-spectrum activity, and was particularly effective against opportunistic yeast pathogens [113]. Brevinin-1BYa has no secondary structure in water, while assuming a flexible helix–hinge–helix motif with C-terminal intramolecular disulfide bridge in membrane-mimicking micelles (Figure 2A–D) [113]. Indeed, in the I-Tasser calculations [114,115] (Figure 2A,B) the unfolded and partially folded structure can be obtained. Notably, residues 1, 4, 5, 7, 8, 10, 11 are potential chlorophylla binding sides, while residues 2, 3, 6, and 7 are nucleic acids binding sides. Nevertheless, the therapeutic potential of brevinin-1BYa is limited by its high hemolytic activity against human erythrocytes (LD_50_ = 10 μM) [116]. [C18S,C24S]brevinin-1BYa (Figure 2E,F) involves substitution of the conserved cysteine residues by serine residues, which results in an acyclic analogue eightfold reduction in hemolytic activity with retained high potency against Gram-positive bacteria, including strains of *S. aureus* MRSA (MIC = 5 μM). However, activities against Gram-negative bacteria and yeast species were reduced [117]. The non-linear relationship between the hydrophobicity of α-helical AMPs and their hemolytic activity is well documented [118]. It is possible that the [C18S,C24S] analogue fails to maintain a non-bonded loop structure akin to that of the native peptide with reduced cysteines as the Ser24 residue is not sufficiently hydrophobic to associate with the largely hydrophobic region of the helix [116]. I-Tasser calculation show that [C18S,C24S] analogue forms a helical structure; nevertheless, the C-end remains unfolded (Figure 2E,F), and the overall 3D structure of the peptide differs from that of brevinin-1BYa.

Snake venom toxins have valuable potential in the design and synthesis of novel antimicrobials. Lys49 phospholipase A2s (PLA2s) are multifunctional snake toxins able to induce a huge variety of therapeutic effects and consequently serve as templates for new drug leads. Almeida et al. synthesized five oligopeptides mimicking regions of the antibacterial Lys49 PLA2 toxin (CoaTx-II isolated from *Crotalus oreganus abyssus* snake venom). The 13 amino acid peptide pC-CoaTxII, corresponding to residues 115-129 of CoaTx-II (KKYRIYPKFLCKK), was able to reproduce the promising bactericidal effect of the toxin against multi-resistant clinical isolates of Gram-negative bacteria (MIC*^P. Aeruginosa^*(μM) = 5.95). Even though, the molecular dynamics (MD) simulations showed that pC-CoaTxII is unstructured [119], the I-Tasser calculation (Figure 3) showed the peptide’s susceptibility to form a helical shape. A possible antimicrobial mechanism of pC-CoaTxII is through strong interaction with anionic lipid membranes such as those in bacteria. In silico studies suggested formation of a water channel across the membrane upon peptide insertion, eventually leading to bacterial cell disruption and death [119].

Tachyplesin-1 (TP1) is a 17 amino acid AMP (KWCFRVCYRGICYRRCR) extracted from the hemocytes of the horseshoe crab *Tachypleus tridentatus* [120]. It forms a β-hairpin structure (Figure 4), both in aqueous solution and in lipid-mimicking environments [121,122], and has two disulfide bridges (Cys3-Cys16, Cys7-Cys12) that play an important role in broad spectrum antimicrobial activity [123]. The reduction in antimicrobial activity, when Cys residues are removed from the peptide’s sequence, is presumably linked to the subsequent loss of β-sheet stacking [124,125,126]. TP1 exhibits potent activity against Gram-positive (MIC*^B. pseudomallei^*(μM) = 62) and Gram-negative (MIC*^E.coli^*(μM) = 22) bacteria, as well as fungi. It did not induce resistance in short-term studies [127], but caused decreased susceptibility under long-term continuous selection conditions [124]. However, TP1 is highly toxic toward mammalian cells (IC_50_^HEPG2^(μM) = 110), making it unsuitable for therapeutic development. Edwards et al. [7] prepared a systematic study of a clear structure−function relationship for each amino acid in the sequence, while modulating charge and hydrophobicity by residue modification and truncation of the peptide. They were able to assess the effects of amino acid replacements on antimicrobial activity, cytotoxicity, and hemolytic activity. Moreover, they evaluated the membrane binding affinity and stability of the most interesting peptides. Three modified peptides: (TP1[F4A]), (TP1[I11A]), and (TP1[C3A,C16A]) maintained the β-hairpin secondary structure motif (Figure 4), and possessed substantially improved therapeutic indexes (26-to 64-fold) over the progenitor peptide, exhibiting MIC*^E.coli^*(μg/mL) = 0.125–4; MIC*^K.pneumoniae^*(μg/mL) = 0.25–16; MIC*^A.baumanii^*(μg/mL) = 0.25–0.5; MIC*^P.aeruginosa^*(μg/mL) = 0.25–2; MIC*^B.subtilis^*(μg/mL) = 0.125–0.25; MIC*^S.aureus^*(μg/mL) = 2–8; MIC*^C.albicans^*(μg/mL) = 2–8; MIC*^C.neofomans^*(μg/mL) = 0.25–4 values and were considerably less hemolyticly toxic. (TP1[F4A]) and (TP1[I11A]), not only conserve the β-hairpin secondary structure motif, but also are highly stable in mouse and human plasma. Of noteworthy mention, I-Tasser calculation showed the presence of potential magnesium and zinc binding sides in TP1 analogues.

The design of a library of peptide antibiotics is usually done using a parent peptide (naturally occurring or designed in the laboratory), followed by structure−activity relationship (SAR) studies and fine-tuning the activity of the peptide [11]. A synthetic antimicrobial peptide library based on the human autophagy 16 polypeptide has been developed by Varnava et al. [128]. The fatty acids conjugation (inspired by lipopeptide antibiotics e.g., polymyxin [129]) and N-Acetylation [130,131,132,133] strategies were used to enhance the activity and serum stability of AMPs. The fine-tuning of structure and length of the fatty acid component of the antimicrobial lipopeptide battacin [134] resulted in peptides library with enhanced potency with respect to the parent Atg1, a human autophagy polypeptide. A 21-residue fragment of Atg16 conjugated to 4-methylhexanoic acid (K30; Figure 5) emerged as the most potent antibacterial agent (MIC*^C.albicans^*(μM) = 30–60; MIC*^P.aeruginosa^*(μM) = 60–120; MIC*^E.coli^*(μM) = 6.4–12.8; MIC*^S.aureus^*(μM) = 0.9–1.8). Moreover, the relationship between the K30 structure (Figure 5) and function in the bacterial membrane was determined. The negatively charged bacterial membrane anchor the K30 peptide, and subsequent interactions with the hydrophobic residues of the peptide, causing the K30 peptide to adopt a helix−loop−helix structure. The folded structure is able to penetrate the nonpolar acyl chain of the lipid molecules, where it subsequently causes membrane disintegration leading to cell lysis [128].

Interference with bacterial virulence is a promising alternative approach or a complementary adjunct to traditional antimicrobial therapy [135]. Antivirulence strategies focus on pathogenicity factors and bypass the pressure on the bacterium to develop resistance. The MgtC membrane protein has been proposed as an attractive antimicrobial target, while it is involved in the ability of several major bacterial pathogens (e.g., *Pseudomonas aeruginosa*) to survive inside macrophages. Moussouni et al. [135] developed an antivirulence strategy targeting MgtC, by taking advantage of a natural antagonistic peptide, MgtR (ILFVADSLQMIPLCLRIWVALKINILFSVL). Heterologous expression of MgtR in *P. aeruginosa* PAO1 was shown to reduce its ability to survive within macrophages, while exogenously added synthetic MgtR peptide lowered the intramacrophage survival of wild-type *P. aeruginosa* PAO1, thus mimicking the phenotype of an MgtC mutant as well as the effect of endogenously produced MgtR peptide [135]. Similar to AMP-mimicking peptides, MgtR has a high susceptibility to form a helical structure. The crystal structure not known at the present time, nevertheless, the I-Tasser structure prediction shows the possibility to form a central helix or a helix–strand–helix structure (Figure 6).

## 7. Amino Acid Conjugated Polymers

Synthetic homo- and co-polymers have low toxicity to host cells and at the same time simulate the functions of AMPs [25,136]. In contrast to AMPs, synthetic antibacterial polymers have a versatile chemical structure, scalability, and lower synthesis costs [137,138]. The polymers are bilateral—on one side cationic residues act with the negatively charged bacterial cell-wall, on the other side hydrophobic domains interact with the lipophilic layers of the bacterial cell-wall leading to the disruption of the cell membrane [28,139,140,141]. Many peptide-mimetic antibacterial polymers are already reported in the literature in recent decades, and are mainly derivatives of polymethacrylates [142,143], polyacrylates [144,145], polynorbornenes [146,147], poly(β-lactam)s [28,139], polymaleimides [148,149], and polycarbonates [150,151]. Extensive studies were dedicated to probing the correlation between antimicrobial activity and polymer amphiphilicity [143,152], structure (random or block) [136,153], type of cationic charge [154], molecular weight [145], and spaced arm (distance from polymer backbone to pendant cationic center) [1].

Systematic optimization of the (co)polymer composition, chain length, hydrophobicity, and cationic charge has generated selected examples that are also highly biocompatible (non-hemolytic and non-cytotoxic in vitro). The extensive review on biomimetic antimicrobial polymers together with polymer chemistry was published by Ergene et al. in 2018 [138], and here, we discuss some recent advances in effective antimicrobial polymer discovery.

Two key parameters are known to affect the efficacy of AMP mimics: molecular weight and amphipathic balance. Polymers of lower molecular weight have greater antimicrobial activity [155], while correct amphipathic balance is necessary to impart both antibacterial activity and selectivity versus bacterial membranes [25]. If the hydrophobicity is too high, selectivity is reduced, and eukaryotic cell death occurs. Researchers increased hydrophobicity through copolymerization of cationic monomers with hydrophobic monomers with alkyl tails of different lengths, which generally resulted in increased bacterial cell death at the expense of selectivity [156]. Multiple polymer backbone structures, including methacrylates [157], β-lactams [158], norbornenes [159], and methacrylamides [160], with varying solubility and inherent polarity have been evaluated as AMP mimics.

Positively charged peptides- lysine and arginine are present in the primary sequence of different AMPs constituting 6–8% incidence of each amino acid [138]. Synthetic polymers containing moieties mimicking lysine and arginine components found in AMPs have been reported to show effectiveness against specific bacteria. Exley at al. [161] created a series of copolymers containing lysine-mimicking aminopropyl methacrylamide (APMA) and arginine-mimicking guanadinopropyl methacrylamide (GPMA) (Figure 7). Copolymers were prepared with varying ratios of the co-monomers (APMA and GPMA), with a degree of polymerization of 35−40 and narrow molecular weight distribution to simulate naturally occurring AMPs. The APMA homopolymer demonstrated the greatest antimicrobial activity against *E. coli*, *S. aureus* and *P. aeruginosa* (MIC(μg/mL) = 500) and the lowest toxicity to mammalian cells. At the same time, the antimicrobial activity of the APMA homopolymer was least affected by changes in salt concentration of both type of the polymers tested [161]. On the contrary, addition of GPMA units in the polymer, lower the antimicrobial activity. Considering MIC ∼100 μg/mL as high antimicrobial activity, and MIC ranging from 500 to 1000 μg/mL as moderate activity, the APMAhomo polymer showed promising results, but needs some future improvements.

Brittin et al. [162] optimized the cationic charged placement, amphiphilic balance and PEGylation content in synthesis of acrylate-based random ternary copolymers comprised of same center cationic, ethyl and poly(oligoethyleneglycol) side chains [162]. The resultant molecules (Figure 8) showed effective antimicrobial activity with low MIC (μg/mL) against *E. coli* and *B. subtilis*, ranging between 22 and 44 (μg/mL) and between 2.5 and 10 (μg/mL), respectively.

Antimicrobial polycarbonates containing primary amino groups can effectively kill bacteria as shown by Nimmagadda et al. [150]. Their single, di-block and random copolymers containing hydrophobic (0–10) and hydrophilic (10–20) units (Figure 9) were revealed to be more effective against different multidrug resistant Gram-positive bacteria strains (MIC*^S.aureus^*(μM) = 5–25; MIC*^S. epidermidis^*(μM) = 5–25; MIC*^E. faecalis^*(μM) = 5–25) respect Maganin II (MIC*^S.aureus^*(μM) = 16; MIC*^S. epidermidis^*(μM) > 50; MIC*^E. faecalis^*(μM) > 50). The amphiphilic units are essential for bacterial killing by bacterial membrane disruption, and random block polymers are more potent than di-block polymers, probably due to stable nanostructures of di-block polymers, which prevent them from interacting with bacterial membranes more effectively.

A new frontier in host defense peptides mimicking is β-peptide polymers. Their potent antimicrobial activity and excellent biocompatibility, together with high stability upon protease associated biodegradation are well ascertained [163,164]. Recently, Chen et al. presented the new potent antimicrobial peptide polymer named 80:20 DM:Bu (Figure 10). It is composed of two subunits, one hydrophilic/cationic subunit (named DM) and one hydrophobic subunit (named Bu), to have both cationic charges and amphiphilicity as synthetic mimics of HDPs [27,28,32]. The polymer displayed fast, potent, and broad-spectrum activities upon multiple multi-drug resistant (MDR) strains of bacteria (MIC*^S. aureus^* (μg/mL) = 12.5; MIC*^s.haemolyticus^* (μg/mL) = 3.13; MIC *^P.aeruginosa^* (μg/mL) = 12.5; MIC*^E.coli^* (μg/mL) = 50), resistant to ampicillin (MIC*^S. aureus^* (μg/mL) > 200; MIC*^s.haemolyticus^* (μg/mL) > 200; MIC *^P.aeruginosa^* (μg/mL) > 200; MIC*^E.coli^* (μg/mL) > 200) and streptomycin (MIC*^S. aureus^* (μg/mL) = 25; MIC*^s.haemolyticus^* (μg/mL) > 200; MIC *^P.aeruginosa^* (μg/mL) > 200; MIC*^E.coli^* (μg/mL) = 100). Moreover, peptide polymer 80:20 DM:Bu exerted satisfactory activity against *K. pneumoniae* (MIC (μg/mL) = 50–100) and *A. baumannii* (MIC (μg/mL) = 12.5–50).

In 2019, Zhou et al. [165] presented a new concept of designing novel biocompatible antibacterial copolymers and expanded the categories of next-generation antibacterial agents without inducing drug resistance. The series of di-block copolymers were inspired by Poly(ε-caprolactone) (PCL) natural AMPs. Poly(ε-caprolactone) (PCL) is a biodegradable copolymer with controlled degradability, biocompatibility, and miscibility with other polymers, properties [166,167,168] excellent for biomedicine applications [166]. PCL (hydrophobic component) was coupled with polylysine (K_n_) to form diblock copolymers named PCL_16_-b-K_n_ (Figure 11). Three polymers PCL_16_-b-K_11_, PCL_16_-b-K_20_, PCL_16_-b-K_27_ showed excellent antimicrobial activity against *E. coli* (MIC (μg/mL) = 8–32) and *S. aureus* (MIC (μg/mL) = 8–16). The membrane disruption mechanism and cytoplasm leakage were observed for both *E. coli* and *S. aureus* treated with PCL_16_-b-K_20_ copolymer. The cytotoxicity tests conducted on human dermal fibroblasts for 72 h showed that even at 500 μg/mL PCL_16_-b-K_20_ concentration (fifty times higher than MIC activity), more than 70% of the normal human cells were still alive [165]. It was also shown that PCL_16_-b-K_n_ copolymer vesicles exhibit the value of a potential application as multifunctional drug-carrier systems with antibacterial capability in cancer therapy [169].

Hydrophobicity modulation through incorporation of amino acids in cationic polymers can provide a significant opportunity to design new amino acid conjugated polymers (ACPs) with potent antibacterial activity and minimum toxicity toward mammalian cells. In 2019, Barman et al. [170] presented a class of ACPs with tunable antibacterial activity through a simple post-polymer-functionalization strategy. Permanent cationic charge was present in every repeating unit (Figure 12), whereby the hydrophobicity of the entire molecule was tuned through incorporation of different amino acids. The amino acid alteration had a strong influence on antibacterial efficacy, and as the amino acid side-chain hydrophobicity was increased, both the antibacterial activity (against broad spectrum of bacteria) and toxicity increased. ACP including a glycine residue (ACP-1 (Gly), Figure 12) showed very good activity (MIC*^A.baumannii^* (μg/mL) = 8−16) against both drug-sensitive and drug-resistant strains, including clinical isolates. Moreover, after 14 continuous passages there was no propensity for bacterial resistance development against this polymer.

Fixing AMPs on surfaces is a new strategy to enhance their stability and increase local concentration and biological availability. Nevertheless, immobilized AMPs have lower ability to interact with multiple targets [171]. A possible solution is attachment of the peptides to the substrate via a spacer, which increases the flexibility and enhances the functional conformation of AMPs, thereby ameliorating the overall antimicrobial effect [172]. Such a strategy is a new horizon in the fabrication of biocoatings. Recently, Acosta et al. [172] proposed an effective antibiofilm coatings based on protein-engineered polymers and antimicrobial peptides for preventing implant-associated infections [172]. The unstructured GL13K (GKIIKLKASLKLL-NH_2_) peptide (Figure 13) was hybridized with elastin-like recombinamers (ELRs) and subsequently tethered to commercially pure titanium discs, and their biological response was successfully tested against the biofilm activity of oral microorganisms (*S. gordonii* and *P. gingivalis*). This innovative protein-engineered molecular platform could be used for immobilization of different AMPs on an ECM-mimicking polymer for the development of antibiofilm and cytocompatible coatings on titanium.

## 8. Mimicking the Structure of AMPs

Peptoids are non-natural, sequence-specific peptidomimetic oligomers based on a protein-like backbone, but with a side chain appendage at the amide nitrogen (Scheme 4) [23].

Synthetic polymers mimicking antimicrobial peptides have huge therapeutics potential; nevertheless, designing an ideal peptoid drug is complicated because there are many variables that influence the activity and function of the oligomer. The most important factors to consider are the cationic charge, conformational constraint via macrocyclization, and hydrophobicity, which are directly correlated with membrane interactions and biological activities in vitro. Peptoid optimization strategies in use are in vitro trial-and-error approaches [173], high-throughput ligand screening on large peptoid libraries [174], and computational approaches with prediction of peptoid efficacy in vitro [175,176,177]. The synchrotron liquid surface X-ray scattering studies of molecule structure–activity relationships have a new impact on studies of the mechanism of action by peptoid antimicrobials, and suggest optimization strategies for future therapeutics based on peptoids [178]. For instance, substitution of lysine to guanidine groups, increases the cationic charge and guides peptide–peptoid chimeras toward phosphatidylglycerol in bacterial membranes [179]. Macrocyclic peptoids reveal superior in vitro efficacy over linear peptoids, with identical monomer composition, against Gram-positive and Gram-negative bacteria [180]. The optimal hydrophobic parameters, balancing membrane specificity and at the same time low cytotoxicity, have been proposed to form helical peptoids [20].

Of noteworthy mention, peptoids adopt a helical structure [181], and are resistant to proteolytic degradation [182]. Helical peptidomimetic oligomers (foldamers) with a structure similar to that of linear, cationic, facially amphipathic helical antibacterial peptides such as magainins (class of AMPs found in the African clawed frog *Xenopus laevis* [183]) have garnered particular interest [181]. For example, certain amphipathic β-peptide helices are comparable to magainins (Figure 14) in antibacterial activity and selectivity [163]. Parameters that may have an influence on activities (oligomer length, charge, hydrophobicity, chirality, secondary structure and side chain nature), the mode of action of these peptide-mimetics and their selectivity for bacteria versus mammalian cells have been scrutinized over the last ten years [20,173,184,185,186].

The structure–activity relationship among 22 cationic amphipathic peptoids was studied by Mojsoska et al. [187]. All studied peptoids had an overall net charge of +4 and were 8 to 9 residues long; however, the hydrophobicity and charge distribution along the abiotic backbone varied. Changes, such as replacing Ntrp with monomers with specific aromatic side chains, rearranging the charge distribution along the peptoid backbone, and shortening the length, significantly impacted the overall hydrophobicity profiles of the peptoids. Peptoids with high hydrophobicity do not always appear as the most potent against *E. coli* and *P. aeruginosa*, while against *S. aureus*, there is a linear relationship between their hydrophobicity and potency. In addition, increased hydrophobicity caused by introducing highly aromatic residues, such as the N-(2,2-diphenylethyl)glycine (Ndpe) monomer, is strongly correlated with a loss of antibacterial specificity, resulting in high toxicity in mammalian cells.

Barron and colleagues showed that the threefold periodicity of the poly-proline type I-like (PPI-like) helix of α-peptoids [188] enabled mimicry of the magainin helical structure [23]. The new amphipathic helix was built by periodic incorporation of a cationic side chain (NLys) every three residues, the remaining positions being occupied by the aromatic a-chiral (S)-phenylethyl side chain (Nspe), which helps to format the helix structure and provides hydrophobic helical faces. The dodecamer (NLys-Nspe-Nspe)_4_ (Figure 15) showed excellent broad-spectrum antibacterial activities (MIC*^E.coli^* (μM) = 3.5–14), but also displayed potent cytotoxicity toward carcinoma cells (LC_50_(breast cancer)(μM) = 5; LC_50_(prostate cancer)(μM) = 5; LC_50_(ovarian cancer)(μM) = 6; LC_50_(fetal lung fibroblast)(μM) = 8; LC_50_(primary dermal fibroblast)(μM) = 8) [189].

Following this mimicking trend, Shyam et al. prepared a series of 1,2,3-triazolium-based cationic amphipathic peptoid oligomers that mimic antimicrobial helical peptides [190]. They explored the potential of the triazolium group as a cationic moiety and helix inducer to develop potent antimicrobial helical peptoids. Several triazolium-based oligomers, even of short length, selectively killed bacteria over mammalian cells. Among tested oligomers H-(Naetm-Nspe-Nspe)_4_-NH_2_ (Figure 16) was the most effective (MIC*^E.coli^* (μM) = 6.3–50; MIC*^E.fecalis^* (μM) = 11; MIC*^S.aureus^* (μM) = 10).

The amino acids—leucine and lysine—have strong helix-promoting abilities [191] and have been used for the construction of prototype α-helical antimicrobial peptides, usually referred to as “LK peptides”. Among them, 14- and 15-mer peptides with amphipathic helix, were found to be the most effective against bacteria [44,192], while shorter peptides were inactive. Attaining a secondary structure and amphipathicity are two determining factors for the antimicrobial activity of the LK peptides. Indeed, Monroc et. al. demonstrated that 4−10-residue-long LK peptides have antimicrobial activity only when cyclized [193], due to structural rigidity, and increased binding to the bacteria membranes. Additionally, the presence of numerous Trp residues stabilize the helical structure [194] and increase the affinity of the naturally occurring AMPs peptide for the membrane [195]. Even single tryptophan substitution at certain positions of inactive fragments of the amphipathic helical AMPs can significantly enhance their antimicrobial activity [195]. Following these instructions, and keeping in mind the role of C-terminal amidation on the activity of AMPs and the prevalence of tryptophan residues in several natural and synthetic AMPs, Pandit et al. prepared short peptides, cheaper in synthesis, and just as effective as long AMPs [196], with P5 peptide (MIC*^E.coli^* (μM) = 10; MIC*^P.aeruginosa^* (μM) = 10; MIC*^K. pneumonie^*(μM) = 10; MIC*^S.typhi^*(μM) = 7.5; MIC*^S.aureus^*(μM) = 50; MIC*^C. albicans^*(μM) = 50; MIC*^C.grubii^*(μM) = 15) slightly more effective than the P4 peptide (MIC*^E.coli^* (μM) = 50; MIC*^P.aeruginosa^* (μM) = 50; MIC*^K. pneumonie^*(μM) = 50; MIC*^S.typhi^*(μM) = 15; MIC*^S.aureus^*(μM) = 50; MIC*^C. albicans^*(μM) = 50; MIC*^C.grubii^*(μM) = 25). Moreover, the peptides P4 (LKWLKKL-NH_2_) and P5 (LRWLRRL-NH_2_) (Figure 17) were found to be non-hemolytic and non-cytotoxic to human cell lines. Surprisingly, these small peptides did not adopt any specific secondary structure in the free or sodium dodecyl sulphate (SDS) micellar bound state, and do not need secondary structures to exhibit antimicrobial activity. Electrostatic interaction is enough for P4 and P5 to attach to the membrane, and its successive deformation and lysis [196].

Cationic antimicrobial poly(α-amino acid)s (APAAs) mimic structures and antimicrobial properties of the AMPs were extensively described in a recent review of Shen et al. [197]. APAAs are easy to synthesize, and have prolonged antimicrobial activity, low cytotoxicity, and enhanced stability to protease degradation. A series of random co-polypeptides with various chain lengths (5 to 200 residues) and hydrophobic contents (1–50 mol%) synthesized by Wyrsta et al. [198] mimic the cationic and amphiphilic nature (Figure 18) of many natural antimicrobial peptides. The most reactive samples have high hydrophobic contents and intermediate chain lengths, and in particular, peptides consisting of Lys/Leu with α-helix conformation exhibit good selectivity and activity to the microbial-mimicked membrane.

Moreover, the surface-initiated N-Carboxyanhydride (NCA) polymerization strategies used for APAA synthesis have also been proposed to generate dense polypeptides brushes on surfaces of a wide range of organic and inorganic substrates, including gold, carbon nanotubes, macroporous polymeric templates, magnetite nanoparticles, and silica nanoparticles [199]. Thus, surface-grafting technologies could be used to fabricate APAAs coatings on biomedical devices, and solve the critical problem of device-related infections [200].

Recently, short peptides combining α-helix and β-turn sequence-motif in a symmetric-end template showed enhanced cell selectivity, antibacterial activity (MIC*^E.coli^*(μM) = 2–32; MIC*^B.pyocyaneum^*(μM) = 2–16; MIC*^S.pullorum^*(μM) = 2–32; MIC*^S.aureus^*(μM) = 8–32; MIC*^S.faecalis^*(μM) = 8–32) and salt-resistance. Two peptides PQ (IHKFWRCRRRFCRWFKHI-NH_2_) and PP (IHKFWRPGRWFKHI-NH_2_) tended to form an α-helical structure upon interacting with a membrane-mimicking environment (Figure 19). According to I-Tasser calculation, PP has a manganese binding site (K3, R6, P7 residues), while PQ has a magnesium binding site (R10, C12 and R13 residues). Moreover, they showed significant cell selectivity toward bacterial cells over eukaryotic cells. Their activities were mostly maintained in the presence of different conditions (salts, serum, heat, and pH), which indicated their stability in vivo [201].

Synthetic peptides with high antibacterial activity and low toxicity can be identified with high accuracy using cheminformatics and machine learning, and without the use of an original template sequence [202]. To achieve that, Fjell et al. [203] used a quantitative structure–activity relationship (QSAR) approach utilizing artificial neural networks (ANN), and built computational models of peptide activity based on data from over 1400 random sequences, biased to contain amino acids believed from substitution analyses to be important for antibacterial activity. In addition, they posed an innovative method of generating candidate peptide sequences using the heuristic evolutionary programming method of genetic algorithms (GA), which provided a large (19-fold) improvement in identification of novel antibacterial peptides. To demonstrate the effectiveness of these techniques in identifying drug candidates, an in silico screening of 100,000 peptides was performed. The 10 most active peptides from this optimization were selected for in vivo study, all having a positive charge of 14, a uniform chain length of 9 amino acids and at least three tryptophan residues [204]. Potent broad spectrum activity was observed for GN-2 (RWKRWWRWI-CONH_2_), GN-4 (RWKKWWRWL-CONH_2_), and GN-6 RKRWWWWFR-CONH_2_ peptides (Figure 20) against *E. coli* (MIC^GN−2^(μL/mL) = 6.2; MIC^GN−4^(μL/mL) = 6.2; MIC^GN−6^(μL/mL) = 12.5), *P. aeruginosa* (MIC^GN−2^(μL/mL) = 3.1; MIC^GN−4^(μL/mL) = 3.1; MIC^GN−6^(μL/mL) = 6.2), and *S. aureus* (MIC^GN−2^(μL/mL) = 3.1; MIC^GN−4^(μL/mL) = 3.1; MIC^GN−6^(μL/mL) = 3.1).

From an AMP database [205] with a total of 2619 AMP sequences, AMPs have an average of +3.2 net positive charges and 32.7 amino acids, with glycine (G), lysine (K), and leucine (L) being the most abundant amino acids, and α-helices and β-sheets being the most common secondary structures. For the development of potent and biocompatible AMPs, α-helical cationic AMPs (αCAMPs) have been the most popular species for investigation [36]. Extensive studies have focused on developing new AMPs by either modifying natural ones or following the strategy of artificial design to produce efficient and cost-effective versions. Among other approaches, one key factor lies in altering amino acid sequences to produce AMPs with different amphiphilicities [206,207,208]. Following this trend, a series of surfactant-like αCAMPs based on the general formula of G(IIKK)_n_I-NH_2_ (n = 2−4, and the AMPs are denoted as G2, G3, and G4, respectively) [209] were synthesized. Among them (Figure 21), G3 is optimal with potent bioactivity (MIC*^E.coli^*(μM) = 8; MIC*^B. subtilis^*(μM) = 2) and low cytotoxicity (IC_50_ (μM) = 15 (HeLa cells); = 25 HL60 cells) to the host mammalian cells. Hydrophobic modifications of G3 have been created by replacing the amino acid residues at the peptide N-terminals or C-terminals or the side chains [210,211]. The high selectivity and associated features are attributed to two design tactics: the use of Ile residues rather than Leu and the perturbation of the hydrophobic face of the helical structure with the insertion of a positively charged Lys residue. Moreover, studies for anticancer applications have demonstrated that the stronger the hydrophobicity—the higher the bioactivity, but also the higher the toxicity [209].

Potent in vitro activity of AMPs often does not translate into in vivo effectiveness due to the interference of the host microenvironment with peptide stability/availability. Hence, mimicking the complex environment found in biofilm-associated infections is essential to predicting the clinical potential of novel AMP-based antimicrobials [212]. The antibiofilm activity of the semisynthetic peptide lin-SB056-1 (KWKIRVRLSA-NH_2_; Figure 22) and its dendrimeric derivative (lin-SB056-1)2-K ([KWKIRVRLSA]_2_-K) against *Pseudomonas aeruginosa* in an in vivo-like three-dimensional lung epithelial cell model, and an in vitro wound model (consisting of an artificial dermis and blood components at physiological levels) gave rewarding results. [212]. When alone, Lin-SB056-1 was moderately effective (MIC*^P.aeruginosa^*(μM) = 38.5) in reducing *P. aeruginosa* biofilm formation in 3D lung epithelial cells, but its antibiofilm activity was significantly increased in association with the chelating agent EDTA. The combination of lin-SB056-1 at 38.5 mM with EDTA (0.3 to 1.25 mM), know metal chelating agent, resulted in the reduction of the initial bacterial inoculum to the limit of detection (10 CFU/mL). The dimeric derivative (lin-SB056-1)_2_-K demonstrated an enhanced biofilm-inhibitory activity (MIC*^P.aeruginosa^*(μM) = 19.25) as compared to both lin-SB056-1 and the lin-SB056-1/EDTA combination, reducing the number of biofilm-associated bacteria up to 3-Log units at concentrations causing less than 20% cell death [212].

## 9. Unnatural Amino Acids

Introduction of fluorinated amino acids or other non-natural amino acids such as α-aminoisobutyric acid (Figure 23) in AMPs were proved to improve their resistance to protease [213,214]. A possible mechanism is that the abnormal structure of AMPs can impede the access of protease to the amide backbone. In addition, many peptidomimetics were introduced in order to avoid the undesired degradation of the α-polypeptides in vivo. For example, Gellman and co-workers designed a series of cationic and amphiphilic β-amino acid oligomers and polyamides to mimic the natural AMPs, which exhibited low hemolysis, high antibacterial activity, and extreme stability [215].

Alkene and maleic anhydride copolymer mimic amphiphilic structure and antimicrobial properties of natural antimicrobial peptides. Szkudlarek et al. used 4-methyl-1-pentene (Figure 23) as a hydrophobic co-monomer (structure similarity with leucine) and maleic anhydride (leaves ample space for further design of the hydrophilic part by means of chemical modification) at constant 1:1 ratio to ensure the hydrophobic and cationic part similar to those in Leu:Lys 1:1 LK-peptides [216]. The C3 copolymer showed rewarding antimicrobial activity against Gram-negative (MIC*^E.coli^*(μg/mL) = 20) and Gram-positive bacteria (MIC*^S.epidermidisi^*(μg/mL) = 20).

Labelling with the paramagnetic amino acid TOAC (Figure 24) is a new strategy in the studies of conformation, dynamics, orientation, and physicochemical properties of AMPs [217]. Unexpectedly, labelling with TOAC increases activity against Gram-positive bacteria, as it was shown for Tritrpticin (TRP3; Figure 24), an AMP against bacteria and fungi [218,219,220,221,222,223]. TRP3 is 13-residues long (VRRFPWWWPFLRR) with a net positive charge of (+4) at physiological pH. It exhibits ion channel-like activity in planar membranes, and permeabilizes bacterial cytoplasmic membranes [218], leading to leakage of the cell contents [222,224]. Recombinational studies showed that replacing both P or residues by A, or removal of P5 led to peptide conformational changes, higher membrane permeabilization, and ultimately, increased antimicrobial activity [222,223,225]. Labelling TRP3 with TOAC (TOAC-VRRFPWWWPFLRR (T1) and VRRF-TOAC-WWWPFLRR (T2)) led to the increase in antibacterial activity against *M. luteus* (MIC^T1^(μM) < 0.38; MIC^T2^(μM) < 0.39) and *E.coli* (MIC^T1^(μM) = 4; MIC^T2^(μM) = 1.3), respect to TRP3 (MIC*^M.luteus^*(μM) = 1.1; MIC*^E.coli^*(μM) = 1.9) TOAC presented a greater freedom of motion at the N-terminus rather than at the internal position. Analogously to TRP3, both TOAC-labelled peptides, showed prominent cation selectivity [217].

Most AMPs targeting membrane bilayers are cationic antimicrobial peptides (CAMPs) [226] with numerous arginine or lysine residues (generally from 2 to 9), which account for the positive net charge. CAMPs have an amphiphilic structure, in which cationic and hydrophobic residues are clustered in different spatial regions. Nevertheless, CAMPs are highly sensitive to proteases. In order to overcome their short half-life, different CAMP-mimicking peptides were protected on N- and C-end by 6-aminohexanoic acid residues (Figure 22). All peptides display high activity toward a broad spectrum of pathogens (MIC*^E.coli^* (μM) = 10, MIC*^P.aeruginosa^* (μM) = 1.25–20; MIC*^B.spizizenii^* (μM) = 10–40; MIC*^S.aureus^* (μM) = 5–40) and have high stability [227].

A clever solution for protease resistance is the replacement of L-amino acids with their D counterparts. This strategy was successfully used by Qui et al. [228] who replaced the amino acid residues or the cationic lysine residue with the corresponding D-amino acids in protonectin (Prt; Figure 25). Protonectin (ILGTILGLLKGL–NH_2_) was originally isolated from the venom of the neotropical social wasp *Agelaia pallipes* [229], and has potent antifungal and antibacterial activity with membrane active action mode [230,231]. Qui et al. showed that both the D-enantiomer of protonectin (D-prt) and D-Lys-protonectin (D-Lys-prt) have strong antimicrobial activity against bacteria (MIC*^S.aureus^*(μM) = 4–128; MIC*^E.coli^*(μM) = 2–8; MIC*^B.subtilis^*(μM) = 8–128; MIC*^S.epidermis^*(μM) = 8–16; MIC*^K.penumoniae^*(μM) = 8–258) and fungi (MIC*^Sakazaii^*(μM) = 8–64). Moreover, D-prt showed strong stability against trypsin, chymotrypsin and the human serum, while D-Lys-prt only showed strong stability against trypsin. D-Lys-prt still kept typical α-helical structure in the membrane mimicking environment, while D-prt showed left hand α-helical structure. In addition, all D-amino acid substituted analogues or partially D-amino acid substituted analogues could act on bacteria with mechanism of protonectin, and disrupt the integrity of membrane while leading to the cell death [228].

## 10. Mimicking Peptide Bonds 

Peptidomimetics can mimic primary, secondary, and even tertiary structures of peptides and proteins, and therefore they have been developed for biomolecular recognition and modulation of protein interactions. In addition, peptidomimetics display advantages over conventional peptides, including resistance to enzymatic hydrolysis, improved bioavailability, and enhanced chemo-diversity [232]. The past decade showed the fast progress in developing biomimetic oligomers, including β-peptides [233] peptoids [234], α-aminoxy-peptides [235], α/β-peptides [236], azapeptides [237], oligoureas [238], aromatic oligoamides [239]. Nonetheless, the development and application of peptidomimetics is still limited due to the availability of backbones and molecular frameworks.

A new class of peptidomimetics, “γ-AApeptides”, based on the chiral γ-PNA backbone was developed by Shi et al. [17]. These γ-AApeptides are oligomers of γ-substituted-N-acylated-N-aminoethyl amino acids resistant to proteolytic degradation and possess the potential to enhance chemo-diversity γ-AApeptides could mimic the primary structure of peptides, as they project the same number of side chains as peptides of the same lengths. Moreover, γ-AApeptides can fold into discrete secondary structures, such as helical and β-turn-like structures, and they can mimic host-defense peptides and display potent and broad-spectrum activity toward a panel of drug resistant bacterial pathogens. Linear γ-AApeptides link amphiphilic building blocks (containing one hydrophobic and one cationic group in each building block) together, and the sequence may adjust the conformation to adopt a globally amphipathic structure on the surface of bacterial membranes. Indeed, γ-AA16 (Figure 26) kills bacteria (MIC*^E.coli, E.faecalis^*(μg/mL) = 5) through the disruption of bacterial membranes. Sulfono-γ-AApeptides form helical structures [240], that mimic cationic host-defense peptides e.g., magainin 2. Among them, γ-AA22 (MIC*^E.coli, E.faecalis^*(μg/mL) = 2) and γ-AA23 (MIC*^E.coli^*(μg/mL) = 4; (MIC*^E.faecalis^*(μg/mL) = 2) (Figure 26) possess excellent antimicrobial activity [241]. Cyclic γ-AApeptides successfully mimic cyclic peptide antibiotics. This is particularly seen for γ-AA24 (MIC*^E.faecalis^*(μg/mL) = 5) (Figure 26), which displays broad-spectrum antimicrobial activity [18].

## 11. Enhancing AMPs Activity with Metal Ions

Enhancement of antibiotic activity through complexation with metal ions is well ascertained. Copper (II), zinc (II) and silver (I) metal adducts with different antibiotics were extensively studied in recent years [242,243,244,245]. Cu(II) and Ag(I) themselves have antimicrobial activity [246], but together with antibiotics e.g., ampicillin, penicillin G and Cefuroxime, they present synergistic effects against Gram-positive (MRSA) bacteria (*S. aureus*). Zn(II) is an essential cofactor for metallo-β-lacamases (enzyme resistant to β-lactan antibiotics), and together with ampicillin and penicillin G shows a high synergistic effect [245].

Not surprisingly, AMPs also bind metal ions (30% of proteins in the living cell coordinate at least one metal ion [247]) and their activity against bacteria can be enhanced upon metal binding. AMPs with a metal chelating ability can simply compete with bacteria for their essential metal ions and lead to the bacteria’s death under starvation. Psoriasin enters bacterial cells and sequesters Zn(II) ions [248], while Microplusin binds copper(II) ions, making them unavailable for the pathogen [249]. Metal binding can also have toxic effects. For instance, copper(II)/Colistin adducts cleave RNA molecules and have dangerous side effects [250].

AMPs change the structure and charge upon metal binding, leading to acquisition and/or enhancement of antibacterial activity [248]. Binding Zn(II) to Calcitermin increases the positive charge and facilitate the interactions with negatively charged bacterial membranes [251]. A similar effect can be observed in zinc(II)/Histidine rich glycoprotein (HRG) abundant in human plasma [252]. Histatins compose a wide group of short, cationic salivary peptides secreted by the parotid and sub-mandibular salivary gland. The C-terminal 16 amino acids fragment is rich in histidine residues, giving an opportunity for complex formation with copper(II) and zinc ions. Indeed, histatin 1 and 3 complexes with metal ions were already established [253], even if the exact functioning of metal adducts is still under study.

Mimicking a metal binding motif is a new strategy in enhancing AMPs activity. One of the most studied strategies is the use of Amino Terminal Copper and Nickel (ATCUN) binding motif. ATCUN is a three-amino acid sequence, finishing with histidine residues (XXH), with high affinity to divalent transition metal ions. The ATCUN motif is present in human albumin, the most abundant metal and drug carrier in plasma [254], but is also found in other metal-binding proteins (e.g., histatins) [255]. Recently, Agbale et al. [256] engineered ATCUN motifs into the native sequence of two AMPs: CM15 and citropin1.1. The incorporation of metal binding motifs changed the antimicrobial activity of the peptides against a panel of carbapenem-resistant enterococci (CRE) bacteria, including carbapenem-resistant *Klebsiella pneumoniae* (KpC+) and *Escherichia coli* (KpC+). The antimicrobial activity was modulated according to the type of ATCUN variant utilized. For instance, CM15 modulated by incorporation of the GGH and VIH ATCUN motifs had 4-fold and 8-fold increased potency against carbapenem-resistant *K. pneumoniae* (KpC+ 1825971), respectively, with respect to the potency of the original peptide. It is noteworthy that there is no need to incorporate Cu(II) metals within the ATCUN-AMP complex drugs since the motif is able to scavenge metal ions in the plasma or target organism. This should avoid many regulatory bottlenecks associated with the use of metals in drugs [257].

Incorporation of mimosine (MIM) residues in the peptide backbone is another rewarding strategy in enhancing antimicrobial activity upon metal binding. Mimosine is a non-protein amino acid with various properties, such as antibacterial, anti-inflammatory, anti-cancer, and anti-virus. Due to its structural similarity with deferiprone (DFP), mimosine is an excellent chelator of Cu(II), Cd(II), Al(III), Fe(III), Ga(III), Gd(III), and In(III) metal ions, as well as actinides and lanthanides. Recent studies showed that in silico design of mimosine-containing peptides is an effective tool in prediction of metal/complexes [258], which increases the antimicrobial activity of free mimosine peptides. For instance, iron(III) complex with 6-amino acid peptide with three MIM residues (H-Mim-Gly-ProGly-Mim-Gly-Gly-Mim-OH) showed 15 times higher activity against *S. aureus* and *B. cereus* strains relative to the free peptide [96].

## 12. Bacteriocins

Bacteriocins comprise a large and functionally diverse family of AMPs produced by a variety of bacteria and play a critical role in mediating microbial interactions and in maintaining microbial diversity [259]. Most Gram-positive bacteria produce bacteriocins, which inhibit the growth of alike or closely related bacterial strains with a narrow spectrum of activity. Some bacteriocins have activity against pathogenic and opportunistic bacteria (including multidrug-resistant species), not discriminating between antibiotic resistant and sensitive strains [260]. Moreover, bacteriocins shown antiviral, antiprotozoal and anticancer activity [261]. Therefore, bacteriocins are considered as alternatives to conventional antibiotics, and represent promising candidates as antibiotic synergists, or alternatives to enhancing the therapeutic effects of current infection treatments and decrease the prevalence of resistant strains.

The term bacteriocins is referred to as a ribosomally-produced peptides composed of 20-60 amino acids, mostly cationic and hydrophobic [262,263]. Initially, bacteriocins were divided into three classes differing in function, molecular weight, amino acid sequence and physicochemical properties. Class I bacteriocins consist of small (<10 kDa) and heat-stable peptides that are post-translationally modified, resulting in the non-standard amino acids, such as lanthionine and β-methyllanthionine [264]. Class II bacteriocins are small (<10 kDa), temperature- and pH-resistant peptides. Subsequently, class II bacteriocins are divided into subclasses based on structure and modifications: subclass IIa bacteriocins (known as pediocin-like bacteriocins, consist of the anti-listerial one-peptide with one or two disulfide bonds, and a conservative N-terminal motif). Subclass IIb bacteriocins are two-peptide bacteriocins, subclass IIc bacteriocins are cyclic bacteriocins. Class III bacteriocins (>30 kDa) are heat-sensitive, protein-like bacteriocins produced by both Gram-positive and Gram-negative bacteria [263,265]. A fourth class of bacteriocins, initially described by Klaenhammer 1993 [266], has been aborted and renamed as bacteriolysins, which comprise large complexes with carbohydrate and lipid moieties such as leuconocin S and lactococcin 27 [267]. Given the complexity and diversity of bacteriocin, as well as the identification of novel ones, the classification scheme is constantly evolving.

Bacteriocins act in different modes, but most of them interact initially with a specific receptor on the target cell and form pores in bacterial cell-membrane, resulting in a dissipation of proton-motive force that leads to cell death [264]. Class I bacteriocins kill target microorganisms by pore formation in the cell membrane and by inhibition of cell-wall peptidoglycan biosynthesis. Class II bacteriocins, such as pediocin and lactococcin, bind to a membrane protein component of the mannose phosphotransferase system to interact with the membranes. The membrane permeabilization allows bacteriocins to enter the cytoplasm, where they destabilize DNA/RNA integrity, protein and cell wall synthesis and enzyme activity.

In some cases, Gram-negative bacteria resist bacteriocins due to the outer membrane, which acts as an effective barrier. In such cases, some metal chelating agents and environmental conditions (acidic pH, high salt concentration, temperature variations) may destabilize the outer membrane and enables bacteriocins to act towards Gram-negative bacteria [268]. The use of bacteriocins in association with chemical compounds or physical treatments extends their activity spectrum on Gram-negative bacteria and counteract the emergence of resistant bacterial strains [269].

Bacteriocins are already used in our daily life. Lactic acid bacteria (LAB) produce bacteriocins applied in food preservation alone, or in combination with other preservation methods (hurdle technology), to increase the shelf-life and the safety of the foods. Nisin, produced by *Lactococcus lactis*, is authorized as a food additive in the EU (E234) under Annex II of Regulation (EC) 1333/2008 for use in several food categories (clotted cream, mascarpone, ripened and processed cheese and cheese products, pasteurized liquid eggs and semolina and tapioca puddings and similar products). Nisin inhibits different Gram-positive pathogens alone, among them are the particularly dangerous *Clostridium botulinum*, *Listeria monocytogenes*, *Staphylococcus aureus*, *Bacillus and Enterococcus.* When used in combination with other antimicrobials, Nisin is also active towards Gram-negative bacteria [270].

Bacteriocins are promising defenses against pathogenic bacteria, but before market introduction, there is still much to know about their biosynthesis and mode of action. Genomics, peptidomics and proteomics are used to understand the molecular mechanistic of their production, regulation, immunity, and mode of action, and help to decrease cytotoxic effects and increase the target selectivity.

## 13. Conclusions

In front of Fleming’s fearsome prophecies about resistant bacteria, we can find consolation in Pasteur’s optimism “If it is a terrifying thought that life is at the mercy of the multiplication of these minute bodies it is a consoling hope that Science will not always remain powerless before such enemies.” The antimicrobial peptides provide new and promising therapeutic approaches, especially in chronic diseases, like oral cavity pathologies, where they can effectively interfere in the early steps of biofilm formation, and at the same time block the inflammatory effects of bacterial toxins. Human antimicrobial peptides are often unable to exhibit toxicity at physiological concentrations, while other organisms’ AMPs express toxicity in the human body. Clever AMPs-mimicking strategies lead to the development of innovative antimicrobial peptides that express antibacterial activity without harmful side effects. In this review, we gathered the most successful AMPs-mimicking strategies of the last decade to be used as a manual for the future synthesis of new peptides.

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
