# Peer review of "The Best Peptidomimetic Strategies to Undercover Antibacterial Peptides"

_ijms, 2020, doi:10.3390/ijms21197349_

Round 1

Reviewer 1 Report

Well represented work, graphics can be a bit improved resolution. Also would be nice to shed light on peptide polymer chemistry 

Author Response

Dear Editor and Reviewers,

first of all, together with all authors I would like to thank you for your valuable comments and suggestions regarding our manuscript, to which we all agree to. Please find in the following brief descriptions of the changes we made in order to improve our paper.

In regard to Reviewer 2 suggestions:

  • graphics can be a bit improved resolution.

The original high-resolution files will be delivered to Editorial office.

  • Also would be nice to shed light on peptide polymer chemistry.

The current version of the review is already lengthy; we advise the readers (in the chapter: “Amino acid conjugated polymers“) the recent review (2018) of Ergene et al. [140], which describe extensively latest advances in molecular design and chemistry of antimicrobial polymers.

In regard to Reviewer 2 suggestions:

  • Page 3, Line 20: nano-carriers...efficiency: Literature needed.

The proper reference was added at the end of the sentence.

  • Page 8, Line 304: The antibacterial data do not show clearly the improvement of the activity for specific bacteria (not the same bacteria are referred).

The proper corrections were made and the MIC results against E. coli and S. aureus are reported for Temporin-SHf, TetraF2W-RR, Horine and Verine-L peptides.

  • Page 11, Line 342: The first parts of the figure (A to D) are not visible.

The margins of the Figure 2 were reduced for better visibility.

  • Two minor typos I noticed: Page 2, Line 52-54 Word "different" is repeated three times, so a synonym would fit better (like various) and page 5, Line 197: two the most --> two of the most.

The phrases were modified.

  • For figures with graphics: Please check for permissions, if needed.

All figures are original.

In regard to Reviewer 3 suggestions:

  • The manuscript lacks paying attention to an immunogenicity and allergic potential of newly developed peptides/peptidomimetics e.g. prediction on the basis of mathematical calculations of allergens similarities (data bases).

Proper discussion was added at the end of “AMPs: Achilles’ heel“ chapter.

  • The authors should also mention two FDA-approved antibacterial peptides (daptomycin, oritavancin). 

At the end of “Introduction” chapter, the number of FDA approved AMPs was updated according to the latest literature (reference added). Moreover, daptomycin and oritavancin were mentioned as proper examples,

Moreover, the extensive English language correction were made.

Best regards

Joanna I. Lachowicz

Reviewer 2 Report

In the manuscript with ID ijms-934267, the authors are presenting an extended overview of the antibacterial peptidomimetic strategies and developments.

The review is well-written and the language is almost flawless. The schemes/figures are great, there are almost no grammar errors or typos, the references are adequate and the amount of information is enormous.

However, there are some minor issues to be addressed before acceptance of the manuscript:

Page 3, Line 20: nano-carriers...efficiency: Literature needed.

Page 8, Line 304: The antibacterial data do not show clearly the improvement of the activity for specific  bacteria (not the same bacteria are referred)

Page 11, Line 342: The first parts of the figure (A to D) are not visible.

Two minor typos I noticed: Page 2, Line 52-54 Word "different" is repeated three times, so a synonym would fit better (like various) and page 5, Line 197: two the most --> two of the most.

For figures with graphics: Please check for permissions, if needed.

In general, the manuscript seems that it would fit better as a chapter in a book. The structure, contents, the way of writing look more suitable for a book chapter rather that a review in a scientific peer-reviewed journal.

Apart from that, I believe that  it can be published after following the abovementioned comments.

Author Response

(The authors gave the same response as above.)

Reviewer 3 Report

The reviewed manuscript entitled “The best peptidomimetic strategies to undercover antibacterial peptides” presents very important issues related to the development of new antibacterial compounds. The manuscript is generally well written and clearly presented. However, the work should be extended to some issues. The manuscript lacks paying attention to an immunogenicity and allergic potential of newly developed peptides/peptidomimetics e.g. prediction on the basis of mathematical calculations of allergens similarities (data bases). The authors should also mention two FDA-approved antibacterial peptides (daptomycin, oritavancin).   

   Author Response

Dear Editor and Reviewers,

first of all, together with all authors I would like to thank you for your valuable comments and suggestions regarding our manuscript, to which we all agree to. Please find in the following brief descriptions of the changes we made in order to improve our paper.

In regard to Reviewer 2 suggestions:

  • graphics can be a bit improved resolution.

The original high-resolution files will be delivered to Editorial office.

  • Also would be nice to shed light on peptide polymer chemistry.

The current version of the review is already lengthy; we advise the readers (in the chapter: “Amino acid conjugated polymers“) the recent review (2018) of Ergene et al. [140], which describe extensively latest advances in molecular design and chemistry of antimicrobial polymers.

In regard to Reviewer 2 suggestions:

  • Page 3, Line 20: nano-carriers...efficiency: Literature needed.

The proper reference was added at the end of the sentence.

  • Page 8, Line 304: The antibacterial data do not show clearly the improvement of the activity for specific bacteria (not the same bacteria are referred).

The proper corrections were made and the MIC results against E. coli and S. aureus are reported for Temporin-SHf, TetraF2W-RR, Horine and Verine-L peptides.

  • Page 11, Line 342: The first parts of the figure (A to D) are not visible.

The margins of the Figure 2 were reduced for better visibility.

  • Two minor typos I noticed: Page 2, Line 52-54 Word "different" is repeated three times, so a synonym would fit better (like various) and page 5, Line 197: two the most --> two of the most.

The phrases were modified.

  • For figures with graphics: Please check for permissions, if needed.

All figures are original.

In regard to Reviewer 3 suggestions:

  • The manuscript lacks paying attention to an immunogenicity and allergic potential of newly developed peptides/peptidomimetics e.g. prediction on the basis of mathematical calculations of allergens similarities (data bases).

Proper discussion was added at the end of “AMPs: Achilles’ heel“ chapter.

  • The authors should also mention two FDA-approved antibacterial peptides (daptomycin, oritavancin). 

At the end of “Introduction” chapter, the number of FDA approved AMPs was updated according to the latest literature (reference added). Moreover, daptomycin and oritavancin were mentioned as proper examples,

Moreover, the extensive English language correction were made.

Best regards

Joanna I. Lachowicz